# Selective Serotonin Reuptake Inhibitors for the Treatment of Depression in Adults with Down Syndrome: A Preliminary Retrospective Chart Review Study

**DOI:** 10.3390/brainsci11091216

**Published:** 2021-09-15

**Authors:** Robyn P. Thom, Michelle L. Palumbo, Claire Thompson, Christopher J. McDougle, Caitlin T. Ravichandran

**Affiliations:** 1Lurie Center for Autism, 1 Maguire Road, Lexington, MA 02421, USA; rthom@mgh.harvard.edu (R.P.T.); mlpalumbo@partners.org (M.L.P.); cthompson44@mgh.harvard.edu (C.T.); cravichandran@mclean.harvard.edu (C.T.R.); 2Massachusetts General Hospital, 55 Fruit Street, Boston, MA 02114, USA; 3Department of Psychiatry, Harvard Medical School, 25 Shattuck Street, Boston, MA 02115, USA; 4Department of Pediatrics, Harvard Medical School, 25 Shattuck Street, Boston, MA 02115, USA; 5McLean Hospital, 115 Mill Street, Belmont, MA 02478, USA

**Keywords:** Down syndrome, depression, selective serotonin reuptake inhibitor

## Abstract

Background: Depression is a common psychiatric comorbidity in individuals with Down syndrome (DS), particularly adults, with an estimated lifetime prevalence of at least 10%. The current literature on the treatment of depression in adults with DS is limited to case series published more than two decades ago, prior to the widespread use of modern antidepressant medications such as selective serotonin reuptake inhibitors (SSRIs). The purpose of this retrospective chart review study was to examine the effectiveness, tolerability, and safety of SSRIs for depression in adults with DS. Methods: Medical records of 11 adults with DS and depression were reviewed. Assignment of scores for severity (S) of symptoms of depression and improvement (I) of symptoms with treatment with an SSRI was made retrospectively using the Clinical Global Impression Scale (CGI). Demographic and clinical characteristics of the study population, SSRI name, dose, and duration of treatment; and adverse effects were also recorded. Results: All 11 patients (7 male, 4 female; mean age = 27.2 years, range 18–46 years) completed a 12-week treatment course with an SSRI. The median duration of time after initiation of the SSRI covered by record review was 2.1 years, with a range of 24 weeks to 6.7 years. Nine of the 11 patients (82%; 95% CI 52%, 95%) were judged responders to SSRIs based on a rating of “much improved” or “very much improved” on the CGI-I after 12 weeks of treatment (median time of follow-up was 14.4 weeks, with a range of 12.0–33.0 weeks). Adverse effects occurred in four patients (36%). The most common adverse effects were daytime sedation and anger. Conclusions: In this preliminary retrospective study, the majority of patients responded to a 12-week course of SSRI treatment and some tolerated long-term use. Controlled studies are needed to further assess the efficacy, tolerability, and safety of SSRIs for the treatment of depression in adults with DS.

## 1. Background 

Down syndrome (DS) or trisomy 21, is a common genetic syndrome, resulting from an extra copy of chromosome 21. According to the Centers for Disease Control and Prevention, about 6000 babies are born with DS each year in the United States [1], and overall, DS occurs in about 1 in 700 live births [2]. Down syndrome commonly includes characteristic physical features, a variable degree of cognitive impairment, and several medical comorbidities. Medical comorbidities commonly associated with DS include congenital heart defects, thyroid disease, gastrointestinal problems, hematological disorders, hearing loss, ocular disorders, and obstructive sleep apnea [3]. In addition to elevated rates of medical comorbidity, individuals with DS have an increased risk of psychiatric disorders compared to the general population [4,5,6]. Prevalence rates of psychiatric comorbidities have been reported to be as high as 38% and 35% in children and adults with DS, respectively [4,5]. Externalizing symptoms, such as oppositionality, impulsivity, and hyperactivity, are more common in children with DS, whereas internalizing symptoms, such as depression, anxiety, and social avoidance, become more prevalent in adolescence and adulthood [7,8,9]. 

Depression is a common comorbidity in adults with DS, with reported prevalence rates ranging from 6–18% [9,10,11,12,13]. A recent study which included 605 individuals with DS from England and Wales demonstrated that 12.4% of younger adults (16–35 years) and 18.4% of older adults (≥36 years) had a history of depression based on medical record review [13]. Females and males with DS had a four- and five-fold increased risk of depression, respectively, compared to the general United Kingdom adult population [13]. In a separate longitudinal cohort study, 134 adolescents and adults with DS (≥16 years) participated in a detailed psychiatric assessment with psychiatrists who had expertise in DS at baseline and two years later [12]. The two-year incidence of a major depressive episode in this study was 5.2% [12]. Adults with DS have several unique risk factors for developing depression compared to the general population, including cognitive impairment [14], reduced serotonin brain tissue concentration in post-mortem studies [15], high prevalence of thyroid disorders [3], and significant emotional stressors related to the transition to adulthood and loss of school-based programming and services. 

Depression is often underrecognized and undertreated in adults with DS. There are several diagnostic challenges related to the inherent communicative and cognitive limitations. While the clinical characteristics of depression in individuals with DS are often similar to those seen in the general population, including sad mood, anhedonia, decreased appetite and weight loss, social withdrawal, reduced speech, low energy, and psychomotor slowing [6], individuals with DS may have difficulty expressing depressive cognitions such as guilt, worthlessness, self-deprecation, or thoughts of suicide [10]. These clinical features may necessitate taking into account behavioral observations and caregiver reports rather than strict application of diagnostic criteria. A retrospective study assessing the clinical features of depression in DS reported that when strict Diagnostic and Statistical Manual of Mental Disorders, Third Edition (DSM-III-R) criteria were applied, only 50% of depressive episodes diagnosed by expert clinicians met the full criteria [10]. This study also demonstrated that depression was frequently misdiagnosed as dementia in individuals with DS and therefore left untreated [10]. In a retrospective study of 42 adults with DS, not all patients with depression received pharmacotherapy, and no patients received a second medication trial if the first was ineffective [10], suggesting undertreatment of a generally treatable psychiatric comorbidity. 

Data on effective treatment approaches for depression in DS are lacking, and systematic studies on the treatment of depression have been highlighted as a critically needed area of research [16]. The published literature on pharmacotherapy for depression in DS is limited to case reports and case series [6]. No systematic studies on the effectiveness, tolerability, and safety of antidepressants in DS have been published. Three case series were published more than 20 years ago, prior to the widespread use of modern antidepressants, reporting the clinical response to tricyclic antidepressants (TCAs) [17,18,19] and one of these case series also reported on three patients’ response to fluoxetine, a selective serotonin reuptake inhibitor (SSRI) [18]. All three patients who received treatment with fluoxetine had a positive response, two of whom had previously failed to respond to TCAs. Of the remaining six patients described in this case series, three other patients responded to TCAs, one responded to a first-generation antipsychotic, and two did not receive pharmacologic treatments. Medication side effects are not reported in any of these case series. 

Selective serotonin reuptake inhibitors are a class of medications which include fluoxetine, paroxetine, sertraline, fluvoxamine, citalopram, and escitalopram. They selectively block the uptake of serotonin and have several Food and Drug Administration (FDA) indications, including for the treatment of depression, anxiety disorders, obsessive-compulsive disorder, and posttraumatic stress disorder. Selective serotonin reuptake inhibitors have largely replaced the TCAs as the first-line treatment for depressive disorders due to similar efficacy, improved tolerability, and a much safer side effect profile [20]. Unlike TCAs, SSRIs are generally nonlethal in overdose, are not associated with cardiac toxicity, and do not lower the seizure threshold. Modern clinical practice guidelines include SSRIs among the first-line medications for the treatment of depression [21]. Tricyclic antidepressants are considered second-line medications, only to be used after the failure of one or more first-line medications [21]. In clinical practice, SSRIs are typically the first class of medications used to treat depression. A recent study demonstrated that SSRIs comprised 93% of first-line medications for depression in primary care [22]. Selective serotonin reuptake inhibitors have a relatively benign side effect profile and are generally well tolerated. The most common side effects include impaired sexual functioning, sleepiness, and weight gain; 25% of patients consider side effects to be either “very bothersome” or “extremely bothersome” [23]. Despite the widespread use of SSRIs in the general population, their use in patients with DS has only been reported in three patients in a single case series [18]. 

This study aims to provide preliminary naturalistic data on whether treatment with SSRIs is effective, tolerable, and safe in reducing depression symptoms in adults with DS. 

## 2. Methods

### 2.1. Study Participants 

Patients potentially eligible for inclusion were identified using the research patient registry for a large hospital network in the Northeastern United States. The medical records of patients with DS, newly initiated on an SSRI (fluoxetine, paroxetine, sertraline, fluvoxamine, citalopram, or escitalopram) at age 18 years or older for the treatment of a depressive disorder (major depressive disorder, dysthymia, persistent depressive disorder, or mood disorder not otherwise specified), who had at least one follow-up visit after SSRI initiation, treated by a psychiatrist at a tertiary care center outpatient neurodevelopmental disorders clinic in the Northeastern United States from 2011–2021 were identified for detailed review. Patients who did not meet these criteria were not included in the study. Routinely collected clinical notes were retrospectively reviewed, extracted, and coded into a RedCap database. The study was approved as an exempt study by the local institutional review board. The depressive disorder diagnosis was made during the course of clinical care by board-certified psychiatrists (MLP and CJM) with expertise in treating adults with developmental disabilities. The diagnosis was corroborated by review of clinical documentation by a second board-certified psychiatrist (RPT).

### 2.2. Outcomes 

Data were collected retrospectively from the medical record. Demographic data, diagnostic information, including severity of intellectual disability, language ability, medical, and psychiatric comorbidities, concomitant medication and non-medication treatments, and symptom changes were collected. If standardized cognitive testing was not available, intellectual ability was clinically determined. Details of the SSRI trial, including medication name, duration of treatment, starting and maximum dosage, and adverse effects, were also collected. 

The severity of the depressive episode was retrospectively determined based upon medical record review of the clinical documentation using the Clinical Global Impression Severity scale (CGI-S) [24] at the time of SSRI initiation and at follow-up (first psychiatric note following 12 weeks of treatment). The CGI-S is a clinician-rated scale with scores ranging from 1 to 7 (1 = not at all ill; 2 = borderline mentally ill; 3 = mildly ill; 4 = moderately ill; 5 = markedly ill; 6 = severely ill; 7 = extremely ill). 

Treatment response was coded on the Clinical Global Impression Improvement scale (CGI-I) anchored to change in depression symptoms over the initial 12 weeks of treatment. The CGI was developed for use in National Institutes of Mental Health-sponsored clinical trials to provide a brief, stand-alone assessment of the clinician’s view of the patient’s global functioning before and after starting a study medication. The CGI has been shown to correlate well with standard research medication efficacy scales across a wide range of psychiatric conditions and the CGI is used in virtually all FDA-regulated psychiatric trials [25]. The CGI-I is a clinician-rated scale with scores ranging from 1 to 7 (1 = very much improved; 2 = much improved; 3 = minimally improved; 4 = no change; 5 = minimally worse; 6 = much worse; 7 = very much worse). The CGI-S and CGI-I ratings were assigned by a board-certified psychiatrist with expertise in treating adults with developmental disabilities who was not the treating psychiatrist (RPT). 

### 2.3. Statistical Analysis

Categorical variables were summarized using frequencies and percentages. Continuous variables were summarized using means, standard deviations (SDs), medians, and ranges. Treatment response was defined using the CGI-I score, with values ≤ 2 (very much improved or much improved) corresponding to response and values ≥ 3 (minimally improved or worse) corresponding to non-response. Patients were classified as having attained remission and recovery if at least three weeks or four months, respectively, of minimal depressive symptoms [26] had been attained at the time of the most recent psychiatry follow-up visit. Ninety-five percent confidence intervals (CIs) for percentages were calculated using Wilson’s method. Time to discontinuation of SSRI medication was characterized using a Kaplan-Meier survival curve plotted using Stata software (version 14). Other data analysis was conducted using SAS software (version 9.4).

## 3. Results

### 3.1. Study Flow

Records of 46 patients were identified for review, of which 11 were included in the study. Figure 1 shows the number of patients successively meeting each of the eligibility criteria. Complete data were collected for all 11 patients included in the study.

### 3.2. Demographic and Clinical Characteristics of the Sample

Demographic and clinical characteristics of the sample at the time of initiation of SSRI treatment are presented in Table 1. Seven of the 11 patients (64%) were male, and 10 (91%) were White. Age ranged from 18 to 46 years. Three (27%) had a comorbid psychiatric diagnosis in addition to the depressive disorder, and all had one or more comorbid medical diagnoses. Five (45%) had a history of treatment with any psychiatric medication, and three (27%) had a history of treatment with an SSRI.

### 3.3. Characteristics of Depressive Episodes and SSRI Treatment

Selective serotonin reuptake inhibitor treatment was initiated between 2013 and 2020. Characteristics of the depressive episodes and treatment are presented in Table 2. Depression severity at the time of SSRI initiation as rated by the CGI-S ranged from mildly ill (CGI-S = 3) to severely ill (CGI-S = 6), with most patients rated as moderately ill (CGI-S = 4). Fluoxetine was the most commonly prescribed SSRI (*n* = 8). Two patients (18%) were prescribed sertraline, and one patient (9%) was prescribed escitalopram. Five patients (45%) were taking one or more concomitant psychiatric medications, and 10 (91%) were receiving one or more non-medication treatments (nine patients participated in a day program, and five patients received individual psychotherapy). Two patients (18%) had depression with psychotic features.

### 3.4. Response to SSRI Treatment

All 11 patients completed at least a 12-week course of SSRI treatment. The median duration of the follow-up period from time of SSRI treatment initiation to the end of the 12-week treatment course was 14.4 weeks, with a range of 12.0–33.0 weeks. Clinical Global Impression ratings before and after 12 weeks of treatment are presented in Figure 2. Based on the CGI-I rating for the soonest psychiatric visit after 12 weeks of treatment, nine of the 11 patients responded (response rate 82%; 95% CI 52%, 95%), with three patients rated as very much improved (CGI-I = 1) and six patients rated as much improved (CGI-I = 2). Eight of the 11 patients (73%) were rated as mildly ill or less at follow-up (CGI-S ≤ 3), with five patients rated as mildly ill (CGI-S = 3), two patients rated as borderline mentally ill (CGI-S = 2), and one patient rated as not at all ill (CGI-S = 1). One of the remaining three patients, taking fluoxetine, was much improved (CGI-I = 2) but still moderately ill (CGI-S = 5); one taking fluoxetine was minimally improved (CGI-I = 3) and still moderately ill (CGI-S = 4); and one taking sertraline was minimally worse (CGI-I = 5) and severely ill (CGI-S = 6). Of the three patients who had previously been prescribed an SSRI, two of the patients were responders (CGI-I = 1 or 2).

Table 3 summarizes the long-term treatment course beyond the initial 12 weeks of treatment with an SSRI. The median duration of time after initiation of the SSRI covered by record review was 2.1 years, with a range of 24 weeks to 6.7 years. Seven patients (64%) had a sustained positive response to the SSRI until the most recent follow-up, and one patient (9%) had a positive response to fluoxetine and was able to discontinue it after three years of stability without a return of depression. Three patients (27%) had an initially positive response to the SSRI but later experienced either behavioral activation or mania with sustained treatment. Of these three patients, two patients (18%) experienced behavioral activation (irritability, anger, and/or agitation) when the SSRI was titrated to a certain dose (escitalopram 15 mg daily and fluoxetine 20 mg daily). The third patient experienced a manic episode after six months of treatment with fluoxetine 15–20 mg daily. Three patients were optimally managed on the SSRI plus an adjunctive medication, either aripiprazole (a second-generation antipsychotic) [*n* = 2], or buspirone (a serotonin-_1A_ partial agonist) [*n* = 1].

Figure 3 presents results on the duration of medication use. Four patients (36%) discontinued use of SSRI medications in the period covered by psychiatric notes in the medical record: three (at 24 weeks, 31 weeks, and 2.1 years, respectively) because of both loss of effectiveness and difficulty tolerating the medication and one (at 3.1 years) because the medication was no longer needed. The duration of medication use at the time of the most recent psychiatric note for the seven patients who remained on medication ranged from 33 weeks to 6.7 years. For the two patients remaining on sertraline, the doses at the time of last follow-up note were 62.5 mg and 125 mg daily. For the five patients remaining on fluoxetine, mean (SD) final dose was 26.8 (24.3) mg per day, with a range of 14.0–70.0 mg per day. Overall, 9/11 patients (82%; 95% CI 52%, 95%) achieved remission (≥3 weeks of minimal depressive symptoms), and 8/11 patients (73%; 95% CI 43%, 90%) achieved recovery (≥4 months of minimal depressive symptoms) based upon the most recent psychiatric follow-up note.

### 3.5. Adverse Effects

Four of the 11 patients (36%; 95% CI 15%, 65%) had adverse effects reported in their psychiatric notes. Daytime sedation and anger were reported for two patients (18%) each, and weight gain, behavioral activation, irritability, anxiety, and mania were reported for one patient (9%) each. There was no indication of increased suicidal thinking, intent, plan, or attempts in any of the psychiatric follow-up notes. 

## 4. Discussion 

This naturalistic, retrospective study evaluated the real-world effectiveness and tolerability of SSRIs for the treatment of depressive disorders in a small sample of adults with DS. This study reports on the largest sample of individuals with DS treated with SSRIs to date. It also offers insight into how SSRIs are used in a long-term naturalistic tertiary care clinical setting. This study informs on several issues that have not been previously reported in adults with DS, including the clinical characteristics of those receiving treatment with SSRIs, the initial 12-week response rate to SSRIs, long-term tolerability of SSRIs, long-term response, and adverse effects. Selective serotonin reuptake inhibitors are the most commonly prescribed first-line medications for depression in the general population. Previous studies of pharmacotherapy for depression in adults with DS have predominantly reported on TCAs, which are now reserved for treatment-refractory depression in modern psychopharmacology due to the potential for life-threatening adverse effects. The results from this study suggest that the treatment of depression in DS with SSRIs was overall well tolerated and safe, and was associated with clinically significant improvement in symptoms of depression in most adults with DS; however, prospective randomized controlled trials are needed to provide conclusive evidence.

In this study, a majority (82%; 95% CI 52%, 95%) of patients responded to SSRI therapy (CGI-I ≤ 2). This response rate is similar to the data reported in a retrospective study conducted by Myers et al. [18]. Myers et al. reviewed the records of 164 adults treated as outpatients from 1979–1989 in the Down Syndrome Program at the Child Development Center at Rhode Island Hospital and identified nine adults with a depressive disorder, three of whom were treated with an SSRI (fluoxetine). All three patients (100%) who received treatment with fluoxetine (mean dosage: 47 mg per day) were deemed responders [18]. No other reports on the use of SSRIs for depression in DS have been published. Data from our study also indicate that the majority of patients achieved long term response, as indicated by the 82% (95% CI 52%, 95%) remission or recovery rate. This high rate of remission or recovery demonstrates the overall treatability of depression in DS, underscoring the importance of accurate case detection and availability of treatment. 

Findings from this study also suggest that psychotic features may be commonly associated with depression in DS. Two of the 11 patients (18%; 95% CI 5%, 48%) in this sample experienced depression with psychotic features. One patient complained of hearing voices, was observed to be responding to internal stimuli, and exhibited disorganized behaviors, such as trying to leave the house in the middle of the night without clothing. The other patient experienced auditory hallucinations of derogatory content, which resulted in distress manifesting as pushing, screaming, and looking for knives as well as paranoid delusions of people breaking into the house. Both patients were successfully treated with a combination of fluoxetine and aripiprazole (a second-generation antipsychotic). Previous literature also supports an elevated rate of depression with psychotic features in DS. A study including 49 adolescents and young adults (13–29 years) treated in specialized psychiatric clinics reported that 8% of patients with DS had a history of depression with psychotic features [11]. Another study comparing the prevalence of obstructive sleep apnea in adults with DS with or without depression reported that 9/28 (32%) patients in the depression group had accompanying psychotic features [27]. 

Patients in this study generally responded to lower dosages of SSRIs than is typically used in the general population [28]. The mean starting dosage of fluoxetine was 4.9 mg daily and the mean maximal dosage was 25.5 mg daily, which approximates the recommended starting dosage of 20 mg daily in the general adult population [28]. This dosing strategy differs from the three cases reported by Myers et al., in which more typical adult dosing was used (optimal dosages of 20 mg, 40 mg, and 80 mg daily) [18]. Especially in light of the observation that two patients (18%) in our study experienced behavioral activation at a certain dosage threshold (fluoxetine 20 mg and escitalopram 15 mg daily) and a third experienced a manic episode after six months of treatment with fluoxetine 15–20 mg daily, the results from this study support a more conservative dosing strategy.

Although SSRIs were generally well tolerated, four of the 11 patients (36%; 95% CI 15%, 65%) had adverse effects reported in their psychiatric notes. The most common adverse effects were daytime sedation and anger which were reported in two patients each. The majority of patients followed tolerated long-term use, with three patients (27%) discontinuing the medication due to both loss of effectiveness and difficulty tolerating the medication related to either behavioral activation (*n* = 2) or mania (*n* = 1). It is of interest that no patients in this sample reported gastrointestinal side effects, which is the most common reason for early discontinuation of SSRIs [20]. The use of lower dosages of SSRIs and a slower titration schedule are known to mitigate SSRI-related side effects and the dosage titration pattern observed in this study may explain why this was observed.

### Limitations

The results are subject to several limitations, including the small sample size, chart review nature of the analysis, which lacks a placebo or control group and standardized rating scales administered at the time of treatment and potential confounding factors associated with the concomitant use of other psychiatric medications and nonmedication treatments. The primary outcome measure of this study was the CGI, anchored to depression. A limitation of using a global rating for treatment response is the inability to determine which symptoms of depression were responsive to SSRIs. Because no standard depression severity rating scales have been used in patients with DS, an overall clinical impression rating scale rated by a single rater with expertise in treating adults with neurodevelopmental disabilities provides preliminary, yet clinically relevant, information to the literature, which can be used as a basis for developing prospective studies to include and assess depression rating scales in this population. The prevalence of side effects reported in this study may be underestimated, as they were not collected in a systematic fashion and patients with DS may be less able to accurately report side effects due to the cognitive and communication limitations associated with the syndrome. Additionally, although the duration of the initial treatment period was pre-determined to be 12 weeks, the median time of clinical follow-up was 14.4 weeks, with a range of 12–33 weeks. While the sample size was limited to 11 patients, this report includes the largest sample of patients with DS treated with SSRIs for depression. In addition, the diagnosis of depressive disorders was based on expert clinical evaluation rather than through the use of standardized assessment tools. The sample may be biased toward individuals with more serious psychopathology, given that it was completed at a tertiary care center and many individuals with DS and milder depression may not be identified and/or referred to psychiatry. Many of these limitations are a function of the retrospective, naturalistic aspects of the study. This study design does offer the advantage of providing insight into the effectiveness, tolerability, and safety of SSRIs in real-world clinical practice. 

## 5. Conclusions

Overall, most adults with DS and depression responded to a 12-week course of SSRI treatment, and some tolerated long-term use. Although seven of the 11 patients in the study had no adverse effects reported in their psychiatric notes, several experienced behavioral activation and one experienced mania. We believe our findings warrant a future prospective randomized placebo-controlled study of SSRIs in adults with DS. 

## Figures and Tables

**Figure 1 brainsci-11-01216-f001:**
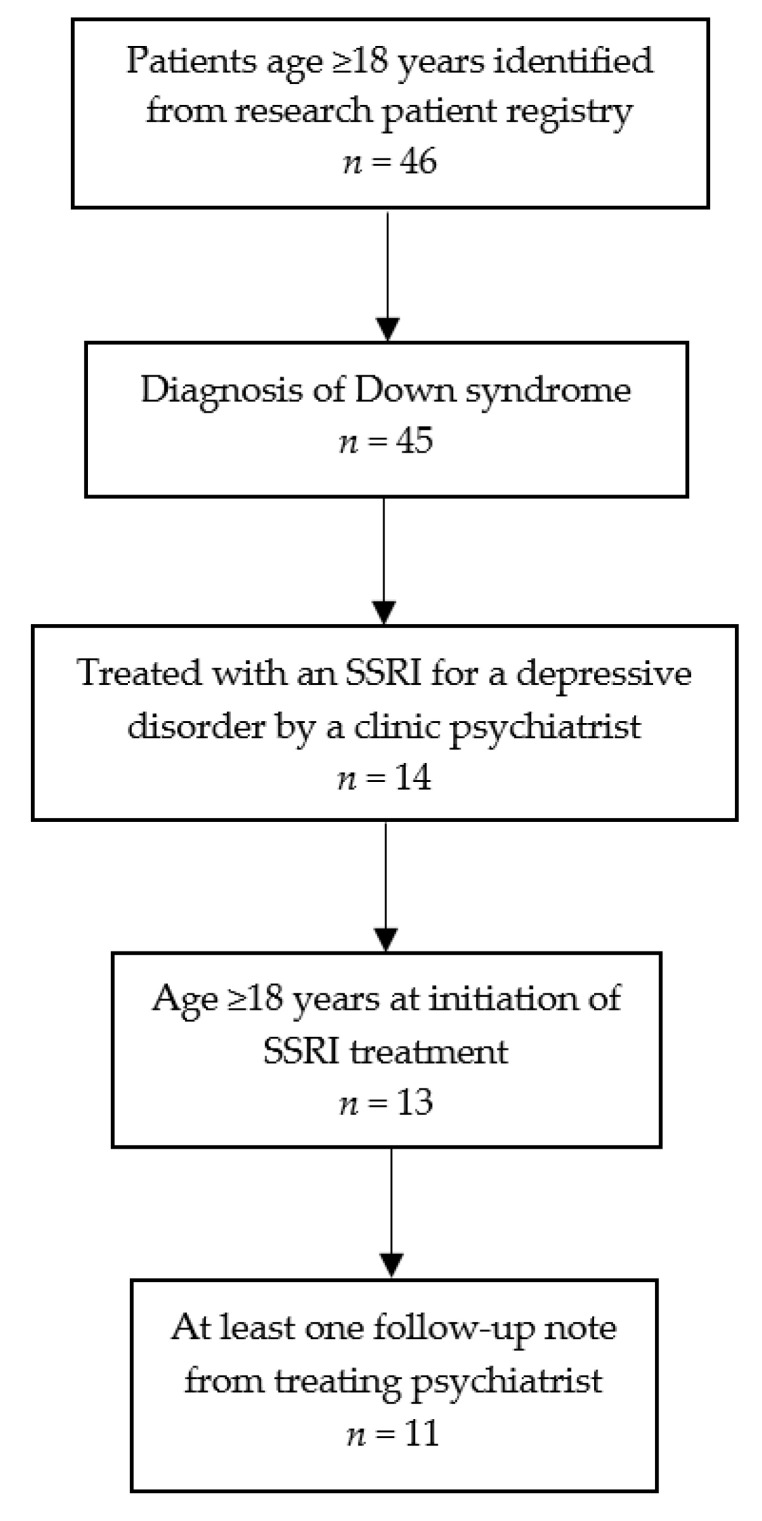
Flow diagram of identification of patients in the study. SSRI: selective serotonin reuptake inhibitor.

**Figure 2 brainsci-11-01216-f002:**
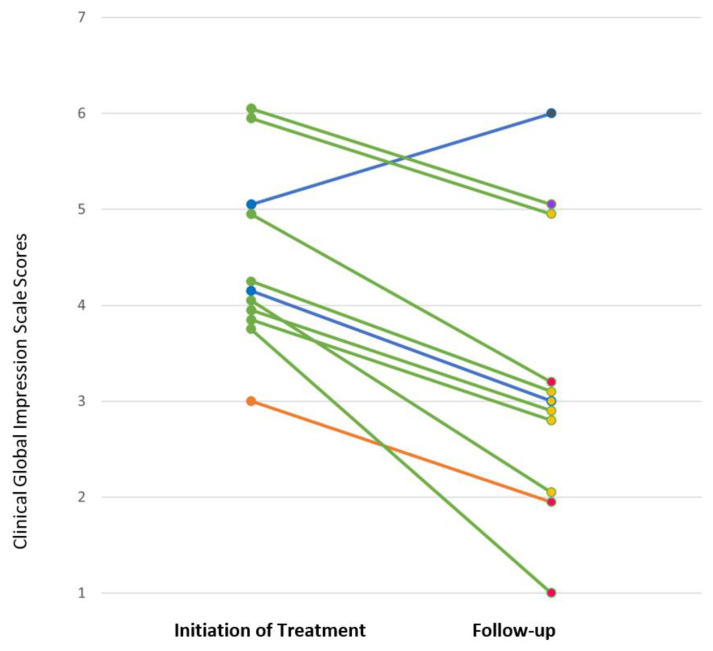
Clinical Global Impression Severity Scale (CGI-S) scores at the initiation of SSRI treatment and at first follow-up visit following 12 weeks of treatment. Each line represents the change in CGI-S score for one patient. Line color corresponds to SSRI: green = fluoxetine, blue = sertraline, orange = escitralopram. Color of marker at follow-up corresponds to CGI-Improvement scale (CGI-I) score: pink = very much improved (CGI-I = 1), yellow = much improved (CGI-I = 2), purple = minimally improved (CGI-I = 3), gray=minimally worse (CGI-I = 5).

**Figure 3 brainsci-11-01216-f003:**
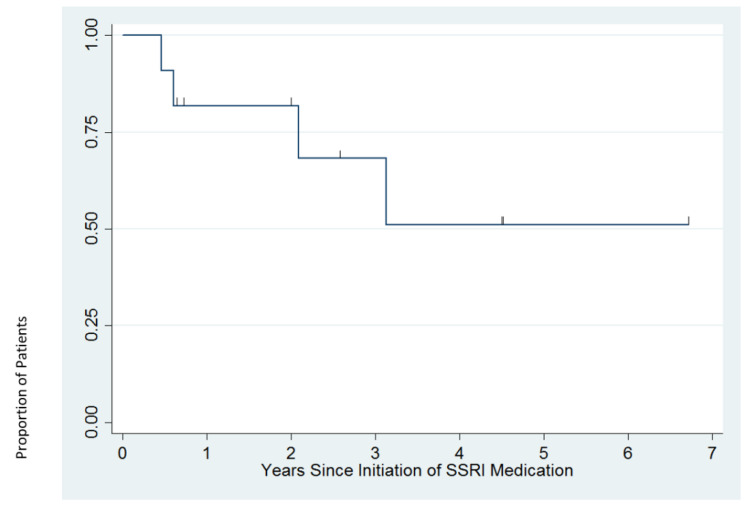
Proportion of patients remaining on SSRI medication over time. Tick-marks correspond to time of last psychiatric note for patients who remained on an SSRI at most recent psychiatric visit.

**Table 1 brainsci-11-01216-t001:** Demographic and Clinical Characteristics of Patients at the Time of SSRI Initiation.

	Eligible for Inclusion *n* = 11
Sex, *n* (%)	
Female	4 (36%)
Male	7 (64%)
Age, mean (SD; range in years)	27.2 (7.7; 18–46)
Race ^1^, *n* (%)	
White	10 (91%)
Other	1 (9%)
Intellectual Disability, *n* (%)	
Any intellectual disability	11 (100%)
Mild	2 (18%)
Moderate	8 (73%)
Severe	1 (9%)
Language Ability	
Full sentences	7 (64%)
Phrase speech	3 (27%)
Single words	1 (9%)
Age at onset of first depressive episode, mean (SD; range in years)	21.6 (10.1; 5–44)
Comorbid psychiatric diagnoses, *n* (%)	
Any diagnosis	3 (27%)
Generalized anxiety disorder	2 (18%)
Alzheimer’s disease	1 (9%)
Comorbid medical diagnoses ^2^, *n* (%)	
Any diagnosis	11 (100%)
Congenital heart disease	7 (64%)
Obstructive sleep apnea	4 (36%)
Hypothyroidism	3 (27%)
Prior psychiatric medications, *n* (%)	
Any prior psychiatric medication	5 (45%)
SSRI ^3^	3 (27%)
Second-generation antipsychotic	2 (18%)
Buspirone	2 (18%)
Donepezil	1 (9%)

^1^ As categorized by the research patient registry; categories include Asian, Black or African American, Hispanic, White, and Other. ^2^ Less frequent comorbid medical diagnoses were constipation (*n* = 2), eczema (*n* = 2), GERD (*n* = 2), hearing loss (*n* = 2), hyperopia (*n* = 2), cataract (*n* = 1), celiac disease (*n* = 1), esotropia (*n* = 1), head trauma (*n* = 1), onychomycosis (*n* = 1), otitis media (*n* = 1), psoriasis (*n* = 1), and type 1 diabetes (*n* = 1). ^3^ SSRI: selective serotonin reuptake inhibitor; GERD: gastroesophageal reflux disease.

**Table 2 brainsci-11-01216-t002:** Characteristics of Depressive Episodes and Treatment.

	Eligible for Inclusion *n* = 11
*Depressive Episode*	
CGI-S score, *n* (%)	
3: Mildly ill	1 (9%)
4: Moderately ill	6 (55%)
5: Markedly ill	2 (18%)
6: Severely ill	2 (18%)
Suicidal ideation at baseline, *n* (%)	1 (9%)
First depressive episode, *n* (%)	7 (64%)
Psychotic features, *n* (%)	2 (18%)
*Treatment*	
SSRI, *n* (%)	
Fluoxetine	8 (73%)
Initial dose in mg, mean (SD; range)	4.9 (0.4; 4.0–5.0)
Maximal dose in mg, mean (SD; range)	25.5 (18.7; 14.0–70.0)
Sertraline ^2^	2 (18%)
Escitalopram ^2^	1 (9%)
Concomitant medication treatments ^1^, *n* (%)	
Any medication treatment	5 (45%)
Second-generation antipsychotic	3 (27%)
Buspirone	2 (18%)
Concomitant non-medication treatments, *n* (%)	
Any non-medication treatment	10 (91%)
Day program	9 (82%)
Individual psychotherapy	5 (45%)

^1^ One patient each was taking an alpha-_2_ agonist, N-acetylcysteine, rivastigmine, and trazodone. ^2^ See Table 3 for dosage information. CGI-S: Clinical Global Impression Severity scale. SSRI: selective serotonin reuptake inhibitor.

**Table 3 brainsci-11-01216-t003:** Summary of 12-week and Long-Term Treatment Course.

Patient	SSRI	CGI-S (Baseline)	CGI-I (12 Weeks)	Follow-up Duration	Long-Term Treatment Course
1	sertraline	5	5	37 weeks	Depression became more severe and psychotic features emerged over first 12 weeks of treatment. Ultimately did well on sertraline 62.5 mg per day and aripiprazole 6 mg per day.
2	sertraline	4	2	6.7 years	Depression continued to improve after 12 weeks with gradual upward dose titration. Doing well on sertraline 125 mg per day at last follow-up.
3	escitalopram	3	1	2.1 years	Depression responded well to escitalopram 12.5 mg per day for ~2 years, but anxiety persisted. Higher doses of escitalopram were associated with fatigue, tearfulness, and irritability. Escitalopram was switched to another SSRI, which also caused tearfulness. The patient was doing well without medications for several months at last follow-up.
4	fluoxetine	5	1	2.0 years	Continued to do well on fluoxetine 14 mg per day at last follow up.
5	fluoxetine	4	2	4.5 years	Continued to do well on fluoxetine 15 mg per day and aripiprazole 2 mg per day at last follow-up. Did not tolerate aripiprazole taper due to irritability.
6	fluoxetine	6	3	2.6 years	Experienced 25–30% improvement on fluoxetine 55 mg per day. Adjunctive buspirone 7.5 mg BID resulted in remission after six months of treatment.
7	fluoxetine	4	2	4.5 years	After six months of fluoxetine 20 mg per day, experienced mania, and fluoxetine was discontinued. Ultimately did well on carbamazepine 300 mg BID and guanfacine 0.5 mg BID.
8	fluoxetine	4	2	24 weeks	Experienced 50% improvement on fluoxetine 20 mg per day. When fluoxetine was increased to 25 mg, experienced agitation, irritability, and anger. Follow-up data after fluoxetine was tapered is not available.
9	fluoxetine	6	2	33 weeks	Experienced moderate improvement on fluoxetine 15 mg per day.
10	fluoxetine	4	2	4.5 years	Depression responded to fluoxetine 20 mg per day. After two years of stability, fluoxetine was tapered and depression recurred. Symptoms resolved when fluoxetine was increased back to 20 mg per day.
11	fluoxetine	4	1	3.1 years	After three years of stability on fluoxetine 30 mg, fluoxetine was tapered and discontinued without return of depression.

SSRI: selective serotonin reuptake inhibitor; CGI-S: Clinical Global Impression Severity scale; CGI-I: Clinical Global Impression Improvement scale.

## Data Availability

The data presented in this study are available in this article.

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
