# Peer review of "Selective Serotonin Reuptake Inhibitors for the Treatment of Depression in Adults with Down Syndrome: A Preliminary Retrospective Chart Review Study"

_brainsci, 2021, doi:10.3390/brainsci11091216_

Round 1
Reviewer 1 Report
In the present study the Authors aimed to provide preliminary naturalistic data on whether treatment with SSRIs might be effective, tolerable, and safe in reducing depressive symptoms in adults with Down Syndrome.
Overall, I found the present naturalistic study quite interesting, well written and timely. However, I have several concerns on it that should be addressed prior publication and these are explained below:
1) As data were collected retrospectively from the medical record, this should be specified in the study title as "a retrospective study". Limitations of such studies must be properly discussed.
2) The title of the paper is "...for the treatment of depression in adults...", but this appears misleading. What it means "depression"? Is it the Major Depression (in such case this must be clearly explained and the inclusion criteria well described) or the depressive symptoms in general (which is surely more common)? This point deserves an accurate revision. Moreover, there are differences in MD and/or depressive symptoms between persons with and without DS? Please, add a comparative table or more informations.
3) My major concern is the lacking of an objective measure of "depression" (see above). I believe that only the CGI doesn't provide a comprehensive overview of this disorder in a special population as persons with DS. This is a major study flaw that should be addressed prior publication. The improvement seen on CGI do not allow us to understand which depressive dimensions gained a beneficial effect by the treatment.
Author Response
In the present study the Authors aimed to provide preliminary naturalistic data on whether treatment with SSRIs might be effective, tolerable, and safe in reducing depressive symptoms in adults with Down Syndrome. Overall, I found the present naturalistic study quite interesting, well written and timely. However, I have several concerns on it that should be addressed prior publication and these are explained below:
1) As data were collected retrospectively from the medical record, this should be specified in the study title as "a retrospective study". Limitations of such studies must be properly discussed.
-The title has been revised to indicate that this was a retrospective chart review study
-The limitations section has been expanded to discuss additional limitations (use of the CGI as the primary outcome measure rather than standardized depression measures, lack of systematic collection of side effects, wide range of initial follow up time).
2) The title of the paper is "...for the treatment of depression in adults...", but this appears misleading. What it means "depression"? Is it the Major Depression (in such case this must be clearly explained and the inclusion criteria well described) or the depressive symptoms in general (which is surely more common)? This point deserves an accurate revision. Moreover, there are differences in MD and/or depressive symptoms between persons with and without DS? Please, add a comparative table or more informations.
-The validity of DSM-5 diagnostic criteria for a major depressive episode is not well established in patients with DS. In fact, a study assessing the clinical features of depression in DS noted that only 50% of depressive episodes in patients with DS met full DSM-III-R criteria for a depressive episode (Cooper, S.A.; Collacott, R.A. Clinical features and diagnostic criteria of depression in Down’s syndrome. Br. J. Psychiatry 1994, 165, 399–403, doi:10.1192/bjp.165.3.399). The conclusion of this study conducted by Cooper et al. was that DSM criteria are “unduly restrictive for this group…Criteria should be modified to facilitate future research.” Because depression may present differently in DS, compared to the general population we did not require a strict DSM-5 diagnosis for a major depressive episode. Rather, patients were included in the study sample if a SSRI was prescribed for the target symptom of depression including a range of depressive disorders (major depressive disorder, dysthymia, persistent depressive disorder, or mood disorder not otherwise specified) which is noted in the first paragraph of the methods section. The diagnosis was made during the course of clinical care by a board-certified psychiatrist with experience treating adults with developmental disabilities and corroborated by review of clinical documentation by a second board-certified psychiatrist.
3) My major concern is the lacking of an objective measure of "depression" (see above). I believe that only the CGI doesn't provide a comprehensive overview of this disorder in a special population as persons with DS. This is a major study flaw that should be addressed prior publication. The improvement seen on CGI do not allow us to understand which depressive dimensions gained a beneficial effect by the treatment.
-Because no standard depression severity rating scales have been used in patients with DS, an overall clinical impression rating scale anchored to depression rated by a single rater with expertise in treating adults with neurodevelopmental disabilities provides preliminary, yet clinically relevant information to the literature which can be used as a basis for developing prospective studies which include and assess depression rating scales in this population. We agree that a major limitation of the CGI is the inability to determine which symptoms of depression are most responsive to treatment.
Reviewer 2 Report
This retrospective case series on antidepressant treatment with SSRIs in 11 patients with Down syndrome has the merit to address a widely under-researched topic.
However, it may not be in the focus of a big part of the very diverse readership of the journal.
Globally, the manuscript is well and clearly written. Relevant aspects of depressive disorder, Down syndrome and pharmacotherapeutic aspects of antidepressant therapy are adequately presented.
Besides the small number of included patients, the main issues are linked to the retrospective, chart-based approach, as indicated in the discussion (limitations section). This issue should however be discussed more thoroughly.
Globally, the methods section should be improved and the limitations be described in more depth.
Comments:
- The title of the manuscript should indicate that the study is retrospective, naturalistic and chart based.
- The methods section should provide additional information:
- According to which diagnostical manual were the diagnosis made?
- Please indicate why the CGI was chosen
- 4, line 205: “Complete data were collected for all 11 patients included in the study.”: please specify the aimed complete data set in the methods section.
- Please indicate eventual exclusion criteria, for example concomitant antidepressant medication, or absence of follow-up after the intial 12-37 week treatment period, or lack of data. Another important precision: was it assessed whether the impaired clinical state of included patients was due to another condition than depression?
- Please indicate why 12 weeks of antidepressant treatment was requested. Finally, the median time of follow-up after 12 weeks was 14.4 weeks, with a range of up to 33 weeks. This is a very wide range
- Results:
- Table 3: Please provide CGI-I scores for the long-term follow-up
- Please indicate how many patients were screened and excluded, and the reason for exclusion. This could give an idea of the quality of data available, the methodology applied or whether there were patients not supporting SSRI treatment after initiation
- Limitations:
- Please discuss the quality of the available data in the charts and whether they seemed to be suitable or not
- Please discuss the issue that the CGI does not specifically asses symptoms of depression, that it is a very broad and quite subjective measure and that there can be many confounding factors. This could also be discussed in the introduction or methods section.
- It has to be stated that results of this study are based on an approximate methodology (retrospective, without any structured assessment of the severity of depression such as questionnaires). The interest in this study is based on the scarcity of available results.
- The assessment of the reported side effects has to be discussed in light of the well known side effect profile of SSRIs. Especially also the absence of other usually prevalent side effects of SSRIs.
Minor comments:
- Abstract line 37: please indicate that results indicated were recorded after a median length of treatment of 14.4. weeks and indicate the range.
- 2, line 51: please indicate the 6000 babies with Down syndrome are born in the US or provide a number for another geographic entity
Author Response
This retrospective case series on antidepressant treatment with SSRIs in 11 patients with Down syndrome has the merit to address a widely under-researched topic. However, it may not be in the focus of a big part of the very diverse readership of the journal. Globally, the manuscript is well and clearly written. Relevant aspects of depressive disorder, Down syndrome and pharmacotherapeutic aspects of antidepressant therapy are adequately presented. Besides the small number of included patients, the main issues are linked to the retrospective, chart-based approach, as indicated in the discussion (limitations section). This issue should however be discussed more thoroughly. Globally, the methods section should be improved and the limitations be described in more depth.
Comments:
The title of the manuscript should indicate that the study is retrospective, naturalistic and chart based.
-The title has been revised to indicate that this was a retrospective chart review study
The methods section should provide additional information:
According to which diagnostical manual were the diagnosis made?
-Because depression may present differently in DS, compared to the general population we did not require a strict DSM-5 diagnosis of depression. Rather, patients were included in the study sample if a SSRI was prescribed for the target symptom of depression including a range of depressive disorders (major depressive disorder, dysthymia, persistent depressive disorder, or mood disorder not otherwise specified) which is noted in the first paragraph of the methods section. The diagnosis was made during the course of clinical care by board-certified psychiatrist and corroborated by review of clinical documentation by a second board-certified psychiatrist.
Please indicate why the CGI was chosen
-We have added the following text about the CGI to the methods section: “The CGI was developed for use in National Institutes of Mental Health-sponsored clinical trials to provide a brief, stand-alone assessment of the clinician’s view of the patient’s global functioning before and after starting a study medication. The CGI has been shown to correlate well with standard research medication efficacy scales across a wide range of psychiatric conditions and the CGI is used in virtually all FDA-regulated psychiatric trials (Busner et al).”
4, line 205: “Complete data were collected for all 11 patients included in the study.”: please specify the aimed complete data set in the methods section.
-The pre-specified data points are listed in the methods section as follows: “Demographic data; diagnostic information including severity of intellectual disability, language ability, medical, and psychiatric comorbidities; concomitant medication and non-medication treatments; and symptom changes were collected. If standardized cognitive testing was not available, intellectual ability was clinically determined. Details of the SSRI trial including medication name; duration of treatment; starting and maximum dosage; and adverse effects were also collected.” Each of these data points were available in the medical record for the 11 study participants.
Please indicate eventual exclusion criteria, for example concomitant antidepressant medication, or absence of follow-up after the intial 12-37 week treatment period, or lack of data. Another important precision: was it assessed whether the impaired clinical state of included patients was due to another condition than depression?
-We have noted that the inclusion criteria include: “patients with DS, newly initiated on a SSRI (fluoxetine, paroxetine, sertraline, fluvoxamine, citalopram, or escitalopram) at age 18 years or older for the treatment of a depressive disorder (major depressive disorder, dysthymia, persistent depressive disorder, or mood disorder not otherwise specified), who had at least one follow up visit after SSRI initiation, treated by a psychiatrist at a tertiary care center outpatient neurodevelopmental disorders clinic in the Northeastern United States from 2011-2021 were identified for detailed review.” We added a clarifying statement on exclusion criteria as follows: “Patients who did not meet these criteria were not included in the study.”
Please indicate why 12 weeks of antidepressant treatment was requested. Finally, the median time of follow-up after 12 weeks was 14.4 weeks, with a range of up to 33 weeks. This is a very wide range
-12 weeks of antidepressant treatment was chosen based on the expected time to achieve a response (Katz MM, Koslow SH, Frazer A. Onset of antidepressant activity: reexamining the structure of depression and multiple actions of drugs. Depress Anxiety. 1996;4(6):257-267). That said, the time to achieve a response from SSRIs remains unknown in DS due to the lack of literature on the pharmacologic treatment of depression in DS. We agree that variability of the time of follow up is a limitation and have added this to the discussion.
Results:
Table 3: Please provide CGI-I scores for the long-term follow-up
-CGI-I scores were not collected for long-term follow up because the primary aim of this study was to assess the initial treatment of depression.
Please indicate how many patients were screened and excluded, and the reason for exclusion. This could give an idea of the quality of data available, the methodology applied or whether there were patients not supporting SSRI treatment after initiation.
-This is detailed in Figure 1.
Please discuss the quality of the available data in the charts and whether they seemed to be suitable or not
-All pre-determined data variables were available in the charts for the 11 patients included in the study.
Please discuss the issue that the CGI does not specifically asses symptoms of depression, that it is a very broad and quite subjective measure and that there can be many confounding factors. This could also be discussed in the introduction or methods section.
-We have noted in the Methods section that the “treatment response was coded on the CGI-I anchored to change in depression symptoms over the initial 12 weeks of treatment.” Further discussion on the limitations of the CGI has been added to the discussion.
It has to be stated that results of this study are based on an approximate methodology (retrospective, without any structured assessment of the severity of depression such as questionnaires). The interest in this study is based on the scarcity of available results.
-We agree that these are limitations to the study, however because of the lack of existing literature on the use of modern antidepressants in DS feel it is an important contribution to the literature.
The assessment of the reported side effects has to be discussed in light of the well known side effect profile of SSRIs. Especially also the absence of other usually prevalent side effects of SSRIs.
-We have added the following text to the discussion: “It is of interest that no patients in this sample reported gastrointestinal side effects, which is the most common reason for early discontinuation of SSRIs [20]. The use of lower dosages of SSRIs and a slower titration schedule are known to mitigate SSRI-related side effects and this dosage titration pattern observed in this study may explain why this was observed.”
Abstract line 37: please indicate that results indicated were recorded after a median length of treatment of 14.4. weeks and indicate the range.
-This has been added.
2, line 51: please indicate the 6000 babies with Down syndrome are born in the US or provide a number for another geographic entity
This has been added.
Round 2
Reviewer 1 Report
The paper is much improved and deserves publication
Reviewer 2 Report
Thank you for taking into account the suggestions of the reviewers. The manuscript is improved significantly.